# The Dynamics of Transaminase and Alkaline Phosphatase Activities in the “Mother–Placenta–Fetus” Functional System

**DOI:** 10.3390/biomedicines13030626

**Published:** 2025-03-04

**Authors:** Elena Kolodkina, Sergey Lytaev

**Affiliations:** Department of Normal Physiology, St. Petersburg State Pediatric Medical University, 194100 Saint Petersburg, Russia

**Keywords:** pregnancy, enzyme level in the blood, aspartate aminotransferase, alanine aminotransferase, alkaline phosphatase, functional system, mother–placenta–fetus

## Abstract

**Background:** The timing of delivery depends on the condition of the fetus and the mother’s body, which is reflected in both the incretion of enzymes in the pregnant woman’s body and their use by the developing fetus in the anabolic process. **The aim:** This work was aimed to analyze the activities of transaminases (aspartate aminotransferase (AST) and alanine aminotransferase (ALT)) and alkaline phosphatase (AlPh) in liquid media (blood serum, amniotic fluid, umbilical cord blood, and placental homogenate) in pregnant women in each trimester of pregnancy and in the postpartum period, considering the timing and type of delivery (term, premature or late delivery, or cesarean section). **Methods:** Data from studies in non-pregnant (n = 45) and pregnant (n = 193) women, including women in labor with different delivery timings (term, premature, and late) and types of delivery (natural birth or cesarean section), were analyzed. Amniotic fluid, umbilical cord blood, and the placental homogenate were collected during labor. The de Ritis coefficient (AST/ALT) was calculated. Alkaline phosphatase activity was determined using the standard constant-time method using Lahema diagnosticum biotests, and transaminase activity was determined using the colorimetric dinitrophenylhydrazine method, according to Reitman and Frenkel. **Outcomes:** The highest alkaline phosphatase activity was recorded in the placenta homogenate (6906.2 ± 208.1 U/mL) in pregnant women who gave birth at term. The highest transaminase activity was found in umbilical cord blood and, in particular, in the placenta in pregnant women with premature and late births. **Conclusions:** The significant role of transaminases and alkaline phosphatase in the transport functions of the histohematic barriers of the mother and fetus was established, which provides a mechanism for the constancy of enzyme levels in blood plasma.

## 1. Introduction

From the standpoint of morphofunctional sciences, blood is a unique tissue that can regulate the flow of metabolites into the physiological systems of the body. It is a liquid medium required for the functioning of numerous enzymes that can be classified not only as indicators of the state or damage of organs but also as biologically active substances that play certain physiological roles [1,2,3,4,5,6,7]. Meanwhile, different metabolic needs require different levels of enzyme activity [8,9]. Blood acts as a medium for regulating metabolic processes and, simultaneously, as a transport, regulatory, and integrating system of the body.

The capacity of enzymatic systems should ensure the rapid renewal of proteins, especially in the liver [10,11]. The rate of aspartate aminotransferase synthesis in the context of a high-protein diet exceeds the rate of liver protein biosynthesis. Hence, the synthesis of alanine aminotransferase is adaptively induced, and its level increases in the blood, enhancing gluconeogenesis [12,13,14]. Evaluating blood enzymes without considering metabolic shifts in the body can lead to inaccurate results. Thus, with a decrease in albumin and total protein and an increase in urea nitrogen, the activity of transaminases increases, and the AST/ALT ratio (de Ritis coefficient) can reach 2.33. Transaminases manifest in the general catabolic direction of protein metabolism, especially during pregnancy [15]. In cases of vitamin D deficiency, a dependence of alkaline phosphatase activity on the phosphorus level in the blood has been recorded [16,17,18].

Blood and other body fluids contain metabolites for various physiological systems, including enzymes. Accordingly, different metabolic needs require different levels of transaminase and alkaline phosphatase activity. Alkaline phosphatase is responsible for the release of glucose from cells and the formation of a phosphate pool. This enzyme is a regulator of transmembrane flows, an indicator of citrate release from bones, and an activator of coagulation processes. AST acts as a metabolism integrator and an indicator of lipid peroxidation, while ALT is a marker of the peripheral catabolic zone. Alkaline phosphatase and aminotransferases assist metabolism in the bodies of the mother and fetus and interact with hydrolases, which are distributed in the “mother–placenta–fetus” system in biological fluids, i.e., the “umbilical cord blood–amniotic fluid–placenta homogenate”. In addition, these enzymes perform a transport function in histohematic barriers, thereby maintaining the constancy of the enzyme composition of the blood, including in pregnant women [19,20]. Figure 1 shows the relationship between the pathways of distribution and synthesis of ALT, AST, and AlPh. The main participants are the heart, liver, bloodstream, and excretory and digestive systems, as well as the fetus and placenta. The peculiarity of these relationships is the site of enzyme synthesis. Transaminases are associated with the heart and liver, while alkaline phosphatase is synthesized in, among other sites, in the “placenta–fetus” system and, separately, in the intestine and bone tissue.

The ontogenetic enzyme level in the blood is the metabolic basis. The fetus has a “sucking” capacity in relation to the maternal organism [20,21]. Thus, calcium in the umbilical cord blood is higher than in the mother, which indicates not only the normal development of the skeleton of the fetus but also the stability of the physiological systems [22,23]. AST and ALT in the tricarboxylic acid cycle (TCA) have two isoforms each and are more stable than lactate dehydrogenase (LDH) and creatine phosphokinase (CPK) [24]. Cholestasis, destructive processes in hepatocytes, or tension in the bile-forming function specifically change the activity of transaminases [25], including during pregnancy [26,27,28,29,30,31,32].

Alkaline phosphatase activity reflects cholestasis and bone tissue damage. Moreover, it is a key indicator of transmembrane processes and the release of glucose from cells into the blood and carries out regulation according to metabolic and enzymological types [33,34,35,36].

There is evidence of a reliable relationship between placental alkaline phosphatase activity and obesity in the mother and child [37], as well as between AlPh activity and the gestational body weight of newborns [38]. In addition, AlPh is involved in the transport of lipids in the intestine and the development of obesity. In this case, placental AlPh is localized in the microvillous plasma membrane of the placental syncytiotrophoblast at the border between the mother and fetus [37]. A positive relationship has been recorded between the activity of AlPh in the plasma membrane of the placental microvilli and the thickness of the mother’s triceps skin fold and the body mass index. The authors concluded that there is an inverse relationship between the expression of the placental alkaline phosphatase gene and the percentage of fat mass in the child, which suggests that placental alkaline phosphatase has a protective function for the fetus against the adverse effects of maternal obesity [37].

Another study revealed a relationship between the activity of placental alkaline phosphatase, biochemical indicators of fetal nutrition (cord blood glucose and albumin), and the height and weight of the newborn [38]. Correlation analysis showed a significant positive relationship between the activity of placental alkaline phosphatase and cord blood glucose, albumin, and the body weight of newborns corresponding to gestational age.

In pregnant women, there is evidence of an association between high-serum alkaline phosphatase levels and adverse pregnancy outcomes, particularly intrahepatic cholestasis and pre-eclampsia [27]. There is a correlation between elevated serum alkaline phosphatase in pregnancy and perinatal complications, which may be used to monitor pregnancies with a high risk of placental damage [39]. It is also believed that an increase in the placental alkaline phosphatase isotype may be a marker of placental insufficiency, preterm labor, or large-for-gestational-age infants. A case of a 30-fold increase in the enzyme in a pregnant woman with full-term delivery and no placental insufficiency has been reported [19].

In this context, the reasons for different levels of enzyme activity, the biological significance of their presence in the blood, and the role of enzymes in the blood containing many substrates remain problematic [40,41].

The aim of this research was to investigate the activities of transaminases and alkaline phosphatase in biological fluids (blood serum, amniotic fluid, umbilical cord blood, and placental homogenate) in pregnant women throughout pregnancy and in the postpartum period, considering the timing and type of delivery.

## 2. Materials and Methods

This study involved 193 Caucasian, primiparous pregnant women aged 18 to 35 years, with a physiological course in the gestation period. They did not have any pathology of the gastrointestinal tract or liver and did not use drugs, alcohol, or nicotine. To select patients, a clinical interview and preliminary questionnaire of pregnant women were conducted to exclude pathologies of organs and systems. The first group consisted of 161 pregnant women whose births ended through the natural birth canal (96 with term births, 34 with premature births, and 31 with late births). The second group consisted of 32 pregnant women whose births ended in emergency surgery (cesarean section).

The gestational age at the time of delivery was determined by the date of last menstruation, the date of the first fetal movement, the first visit to the antenatal clinic, and data from external obstetric and ultrasound examinations.

The clinical data on the course of pregnancy and childbirth were obtained based on the exchange and notification card for the observation of the pregnant woman and the woman in labor, the history of childbirth, and the history of the development of the newborn.

The control group consisted of 45 practically healthy non-pregnant women aged 18 to 30 years (average: 24.2 ± 0.3 years) without any pathologies or concomitant diseases.

This study was conducted on women in the St. Petersburg Snegirev’s Maternity Hospital No. 6. All women (the control and research groups) were informed about the purpose and methods and gave written voluntary informed consent to participate in the study (protocol no. 0608-23 dated 7 August 2023, of the Local Ethics Committee of the Almazov National Medical Research Center).

The content and activity of alkaline phosphatase and transaminases in the peripheral blood sera of non-pregnant and pregnant women were assessed in each trimester of pregnancy and in the postpartum period. Amniotic fluid, umbilical cord blood, and the placental homogenate were collected during the childbirth period. The de Ritis coefficient (AST/ALT) was calculated.

The research design (Figure 2) consisted of the following stages: blood serum was tested once in healthy subjects (women) and four times in pregnant women, regardless of the type of delivery in each trimester of pregnancy and after the birth of the child. The activities of alkaline phosphatase and transaminases were determined in the blood sera of pregnant women at 12–13, 25–26, and 39–40 weeks of pregnancy and once in individuals in the control group. Accordingly, the umbilical cord fluid, amniotic fluid, and placenta were collected from pregnant women. The transaminase (ALT and AST) and alkaline phosphatase activities were assessed in all fluids.

Amniotic fluid was collected during an amniotomy performed to induce labor in women at full-term pregnancy due to a tendency for post-term pregnancy. In premature births, the material was collected in a tray when the fluid was released. In urgent births, it was collected during a timely amniotomy when the opening was 6–8 cm or an early amniotomy when the cervix was 3–4 cm due to the development of the weakness of labor activity. In addition, amniotic fluid was collected during an emergency cesarean section due to the persistent weakness of labor activity, a clinically narrow pelvis, the presence of a large fetus, and acute fetal hypoxia.

AlPh activity was determined with the standard constant-time method using the “Lahema diagnosticum” biotests. The principle is that alkaline phosphates break down 4-nitrophenyl phosphate in the N-methyl-glucamine buffer to form 4-nitrophenol and phosphate. The measure of the catalytic concentration of the enzyme was the amount of 4-nitrophenol, which was determined photometrically using the constant-time method after the enzymatic reaction ended with an alkaline phosphatase inhibitor, which blocks the active center of the enzyme [6].

The activities of transaminases (AST and ALT) were determined using the colorimetric dinitrophenylhydrazine method, according to Reitman and Frenkel [6,37]. AST was determined by measuring the optical density of 2-oxoglutaric and pyruvic acid hydrazones in an alkaline medium; ALT was determined by measuring the optical concentrations of 2-oxoglutaric and pyruvic acid hydrazones in an alkaline medium.

The enzyme levels in non-pregnant women were taken as controls to show the differences from those in pregnant women. The non-pregnant women were a group of practically healthy women aged 18 to 35 years, without any pathologies of the gastrointestinal tract, undergoing planned screening studies. First, the distribution was checked for normality using the Pearson criterion. The average values (M) and their standard error (m) and standard deviation (σ) were calculated. The dependence between the features was assessed using the pair correlation coefficient (r), its error (mr), and the level of significance of differences (according to Student’s t-criterion). The dependence was considered strong when |r| > 0.7 (average) if the modulus of the pair correlation value was within 0.3–0.7. A correlation value of less than the modal value of 0.3 was considered weak.

The nonparametric Mann–Whitney criterion was used when a non-normal distribution was confirmed. Differences between groups in the levels of the studied characteristics were estimated with the nonparametric Mann–Whitney method using the statistical package SPSS 11.0. Differences were considered significant at an error probability of *p* < 0.05. Significance levels of *p* < 0.01 and *p* < 0.001 were also distinguished. The results were statistically processed using Microsoft Excel 2003 spreadsheets and the SPSS 23.0 and Primer of Biostatistics 4.03 programs.

## 3. Results

To study the pathways and mechanisms of the homeostasis of hydrolases secreted by digestive glands (pepsinogen, amylase, lipase, and intestinal phosphatase) during pregnancy, it is useful to monitor enzymes (alkaline phosphatase and transaminases (AST and ALT)) that are involved in transport and metabolic processes in the “mother–placenta–fetus” functional system and which reflect cytolysis in organs and tissues according to their serum levels [1,4].

The timing of childbirth depends on the condition of the fetus and the mother’s body [8,9]. This affects both the incretion of hydrolases in the body of the pregnant woman and their use by the developing fetus in the anabolism of substances and digestive processes [40]. Since the solution to the problem of this study is associated with several height and weight indicators of the mother, these data are detailed in Table 1.

The total weight gain during pregnancy in the women studied averaged 9.3 ± 0.4 kg. The mass–height index in the women studied was 331.2 ± 15.8 g/cm. Most children (90.7%) born to these women had body weight within the normal range (3367.5 ± 75.1 g), with an Apgar score of 7.8 ± 0.3 points in the 1st min and 8.4 ± 0.4 points in the 5th min.

The following indicators were identified in pregnant women with premature births: the body weight before pregnancy was 61.8 ± 2.4 kg, and the height was 164.6 ± 9.5 cm. The total weight gain during pregnancy was 10.3 ± 0.7 kg. The mass–height index was 307.3 ± 11.9 g/cm. In 64.7% of cases, children were born with insufficient weight (average: 2454.1 ± 52.6 g), with an Apgar score of 6.4 ± 0.2 points in the 1st min and 7.2 ± 0.3 points in the 5th min.

In pregnant women with late labor, the body weight before pregnancy was 81.1 ± 5.6 kg with a height of 168.5 ± 11.2 cm. During pregnancy, the total weight gain averaged 12.6 ± 0.6 kg. The mass–height index in the study group of women averaged 394.1 ± 18.3 g/cm. In most cases (77.4%), pregnant women with late labor gave birth to babies weighing an average of 3515.2 ± 63.1 g, with an Apgar score of 7.3 ± 0.3 points in the 1st min and 8.5 ± 0.5 points in the 5th min.

In pregnant women whose labor ended with a cesarean section, their pre-pregnancy body weight was 71.1 ± 3.9 kg with an average height of 166.8 ± 10.4 cm. The total weight gain during pregnancy in the women studied corresponded to an average of 11.8 ± 0.7 kg. The mass–height index was 426.5 ± 25.1 g/cm.

In total, 81.3% of children born to these women had body weight within the normal range (3586.1 ± 66.2 g), with an Apgar score of 7.4 ± 0.3 points in the 1st min and 8.1 ± 0.4 points in the 5th min.

The following patterns of change in AlPh and transaminases were observed in women with timely childbirth (Figure 3): Compared with non-pregnant women (the control group), the alkaline phosphatase activity increased from trimester to trimester during pregnancy and, after childbirth, decreased to values close to those of the control group. The alkaline phosphatase content in umbilical cord blood was equal to that in women’s sera in the third trimester of pregnancy. The highest enzyme activity was recorded in the placental homogenate (6906.2 ± 208.1 U/mL).

The de Ritis coefficient (AST/ALT), an indicator of the adaptation of metabolic flows, was equal to 1.26 ± 0.04 in the control and was reduced to less than one due to a greater increase in the ALT enzyme, which is included in the glucose–alanine shunt and catabolism (Figure 4).

In women with preterm labor, the alkaline phosphatase activity was slightly lower, and transaminase levels were greater than those in pregnant women who gave birth on time (Figure 4).

The transaminase activity at the end of pregnancy approached the baseline level (non-pregnant women). The enzyme levels in the blood were less pronounced in the umbilical cord blood and amniotic fluid, although the dynamics of activity changes were similar to those in pregnant women who gave birth on time (Figure 5).

The transaminase activity in the blood serum of pregnant women with premature births increased from trimester to trimester. Thus, the AST activity exceeded the control data by 1.5 times in the first trimester of pregnancy, by 1.8 times in the second (*p* < 0.05), and by 2 times in the third trimester (*p* < 0.001). It decreased in the postpartum period, reaching the values recorded in non-pregnant women (Figure 6).

The most noticeable changes in ALT activity were during pregnancy and in the postpartum period. Thus, the indicators exceeded the control values by 2–2.2 times during pregnancy (*p* < 0.05) and by 1.3 times in the postpartum period (*p* < 0.05). The de Ritis coefficient in the second and third trimesters exceeded one, whereas it was close to one in the first trimester and after childbirth.

Thus, in preterm labor, the enzymemia of AST, ALT, and alkaline phosphatase ensured the carbohydrate activity of substrates to a greater extent.

In pregnant women with late childbirth, the dynamics of the enzymatic activity of alkaline phosphatase and transaminases persisted, although the amniotic fluid had a higher content of the first enzyme (Figure 7; Table 2).

The dynamics of change in transaminase and alkaline phosphatase activities during pregnancy and childbirth remained constant during the cesarean section (Figure 8). However, the increase in alkaline phosphatase activity was greater than that observed during pregnancy ending in term labor.

The transaminase activity in the blood serum of this studied group of women during pregnancy increased, especially compared with AST (Table 3). The values of the de Ritis coefficient for the trimesters of pregnancy and after childbirth were always higher than one, with almost the same AST and ALT contents, which indicates the adaptation of metabolic processes.

The alkaline phosphatase activity in the placental homogenate in women who delivered at term was 6906.2 ± 208.1 U/mL (Table 4). The enzyme activity in women with premature deliveries was slightly lower (1.1 times; *p* = 0.052) compared to women with term deliveries, while it was 1.1 times higher in women who delivered via cesarean section and those with late deliveries (*p* = 0.068). When studying the transaminase activity of the placental homogenate in groups of women with different terms and types of delivery, the following changes in the indicators were revealed (Table 5).

The AST activity in the placental homogenate of women who delivered at term was 62.31 ± 4.30 U/mL. The enzyme activity was 1.5 times (*p* = 0.045) lower in premature births than in full-term births. An increase in AST activity was observed in late births and births via cesarean section compared with full-term births (1.3 times (*p* = 0.086) and 1.2 times (*p* = 0.037), respectively). There was a weak relationship between AST activity in the placental homogenate of women who delivered at term and via cesarean section (r = +0.24 ± 0.07; *p* = 0.054). The pattern of change in the ALT activity in this biofluid in the studied groups of women was the same as that for AST. The difference was a decrease in ALT activity in women who gave birth via cesarean section (1.2 times compared with women who gave birth at term).

Accordingly, the de Ritis coefficient changed, which was below one in the groups of women who gave birth on time and prematurely and above one in the groups of women with late births and births via cesarean section. Thus, the placenta deposits and concentrates enzymes coming from the mother’s body, the reserve of which is used by the fetus for autolytic digestion. The alkaline phosphatase content in the umbilical cord blood was 6–7 times lower than the activity index of the placenta homogenate. The transaminase activity of the umbilical cord blood in the studied groups of pregnant women had the indices detailed in Table 6.

Women who delivered at term and women who delivered via cesarean section had the highest AST and ALT levels in the cord blood. The AST and ALT activities were 1.3 and 1.6 times lower (*p* = 0.044 and *p* = 0.0001, respectively) in women with preterm deliveries compared with pregnant women who delivered at term, respectively. The same dynamics of change in AST and ALT levels in the cord blood were observed in late deliveries (the AST and ALT activities were 1.1 and 1.5 times lower than in term deliveries, respectively).

Thus, the relationship between the fetus and the mother is ensured by the work of the uteroplacental barrier, along which transaminases and alkaline phosphatases enter the fetal blood serum from the mother’s body.

## 4. Discussion

To study the pathways and mechanisms of the homeostasis of hydrolases secreted by digestive glands (pepsinogen, amylase, lipase, intestinal phosphatase) during pregnancy, it is useful to monitor enzymes (alkaline phosphatase and transaminases (AST and ALT)) that are involved in the transport and metabolic processes in the “mother–placenta–fetus” system and which reflect cytolysis in organs and tissues according to their serum levels [1,4].

The timing of delivery depends on the condition of the fetus and the mother’s body [8,9]. This affects both the incretion of hydrolases in the pregnant woman’s body and their use by the developing fetus in the anabolism of substances and digestive processes [19]. Blood regulates the intake of metabolites into the body’s system and is a medium for the functions of enzymes, including transaminases and alkaline phosphatase [7].

The amniotic fluid occupies an intermediate position between the umbilical cord blood and placental homogenate, which characterizes the transport functions of this enzyme and energy supply in the mother–placenta–fetus functional system [1,41].

The shift in metabolism in the “mother–placenta–fetus” functional system was judged by monitoring transaminases, the activity of which differed from the baseline levels in the control. The activities of both AST and ALT increased in the second and third trimesters of pregnancy. The de Ritis coefficient, an indicator of the adaptation of metabolic fluxes that reflects the integration of protein metabolism, was in the range of 0.9, regardless of the increase in the activity of these enzymes. Only in non-pregnant women was it higher than 1, with an average of 1.27.

Enzymological changes in the blood reflect metabolic rather than diagnostic significance. Alkaline phosphatase is responsible for the release of glucose from cells and the formation of a phosphate pool. These components interact with buffer systems, which are markers of ontogenetic maturity, regulators of transmembrane exchange, indicators of citrate release from bones, and activators of coagulation [32].

Transaminases, at their optimal ratio, regulate metabolic processes as follows:-AST is an integrator of metabolism, a marker of the central zone of catabolism, and an inhibitor of lipid peroxidation;-ALT is a component of the glucose–alanine shunt and a marker of the peripheral zone of catabolism;-AST/ALT is an indicator of the adaptation of metabolic flows [6,10].

All these enzymes ensure metabolism in the body of the mother and fetus and interact with hydrolases distributed in the “mother–placenta–fetus” system in biological fluids: the umbilical cord blood–amniotic fluid–placental homogenate.

According to the literature, different types of metabolism (protein, lipid, and carbohydrate metabolism) are evaluated based on blood enzymes [2,4,40]. The activity of enzymes in the blood is influenced by their ability to persist in free and fixed (sorbed) structures of shaped elements and the vascular endothelium on membranes, enzyme desorption, and their connection with plasma or serum transport proteins [5].

Blood transaminases reflect the manifestation of cytolysis. Meanwhile, programmed cell death in a healthy body is controlled and occurs via apoptosis while preserving the integrity of the cytoplasmic membrane and forming apoptotic bodies without releasing other substances into the surrounding tissues [7,10].

The blood contains numerous proteases, which actively destroy transaminases and other enzymes. Some enzymes are involved in the formation of various complexes that contribute to the course of various biochemical reactions. They are modified while maintaining their basic properties and the ability to function in plasma [7].

Increased alkaline phosphatase activity reflects cholestasis and damage to bone tissue and liver and is an important indicator of transmembrane processes and glucose release from cells into the blood [35].

In particular, there is evidence that newborns born to women with metabolic disorders have increased alkaline phosphatase activity in the umbilical cord blood [12]. AlPh is necessary to compensate for the decrease in phosphorus levels via dephosphorylation. It carries out regulation according to metabolic and enzymological types [16,42].

Researchers have revealed that the activity of blood enzymes depends on the characteristics and type of nutrition [18,43]. It has been shown that amniotrophic and lactotrophic nutrition is provided by enzymes secreted in the female body as well as myhydrolases, which are inversely correlated with transaminases [18,41,44].

According to another study, a high-protein diet increases the activity of enzymes of nitrogen metabolism in the liver, and protein depletion reduces the pool of free amino acids and the activity of the corresponding enzymes [7].

Transaminases not only bind streams of amino acids and proteins but also carbohydrates. Metabolism is integrated in the liver with the participation of transaminases and hydrolases [3].

We conducted a study on the activity of enzymes (alkaline phosphatase and transaminases) in non-pregnant and pregnant women, considering the timing of the delivery and the type of delivery. When studying alkaline phosphatase activity in the blood serum of pregnant women with different types and terms of delivery, we found that the enzyme activity was higher than the baseline control parameters in all the studied groups, with the greatest changes taking place in pregnant women who gave birth on time.

Transaminases were more active in umbilical cord blood and, especially, in the placenta. This was not accidental since these enzymes, at their optimal ratio, regulate metabolic processes in the uteroplacental barrier and its permeability, ensuring fetal–maternal relations [8,9].

In premature and delayed childbirth, the enzyme levels in the blood of AST, ALT, and alkaline phosphatase provide the carbohydrate activity of substrates to a greater extent, as evidenced by their higher transaminase activity compared with that in pregnant women who gave birth on time. Thus, alkaline phosphatase and transaminases play a significant role in the transport functions of histohematic barriers, which ensures the mechanism for the constancy of the enzyme composition of the blood plasma.

## 5. Conclusions

The fetal–maternal relationship is ensured by the stabilization of the uteroplacental barrier, which is determined by the enzyme levels in the blood in relation to transaminases and alkaline phosphatase, which change the levels of metabolite flows and biologically active substances, including digestive hydrolases. Under these conditions, fetal development depends on the intake of nutrients and their effective use in the trophosystems of pregnant women.

An increase in alkaline phosphatase activity from trimester to trimester was noted during pregnancy, and after delivery, it decreased to values close to those of the control. The AlPh content in the umbilical cord blood was equal to that in women’s blood serum in the third trimester of pregnancy. The highest enzyme activity was recorded in the placental homogenate. Alkaline phosphates are widespread in the tissues and fluids of temporary structures formed during pregnancy—in the placenta and amniotic fluid. The enzyme has intestinal, liver, and bone isoforms that participate in transport and energy processes that are most intense during pregnancy.

Transaminases (ALT and AST) reflect energy metabolism, and their connection with alkaline phosphatase ensures transmembrane processes and transport through histohematic barriers, which is important to consider when characterizing the uteroplacental barrier. Transaminase activity reflects the participation of AST and ALT together with alkaline phosphatase in cytolysis and the transport functions of the uteroplacental, fetal, and liver barriers. Transaminase activity was determined in all the tissues and biofluids studied, with the highest content in the placenta during late labor and umbilical cord blood during full-term labor.

## Figures and Tables

**Figure 1 biomedicines-13-00626-f001:**
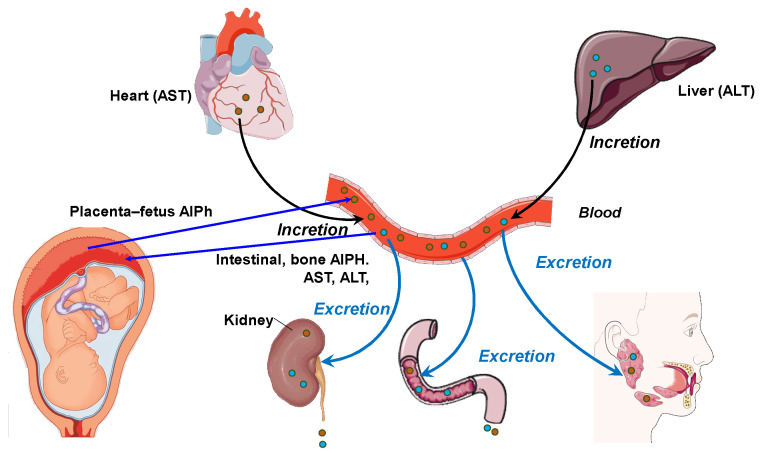
The relationship between the pathways and organs of secretion and incretion of ALT, AST, and AlPh (original).

**Figure 2 biomedicines-13-00626-f002:**
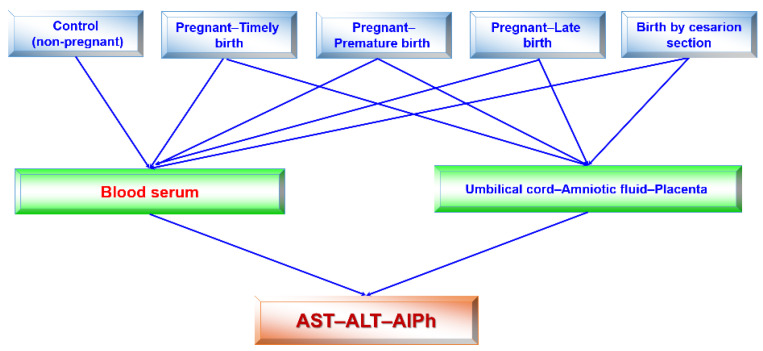
Flow chart (design) of this study. Studies of blood sera in pregnant women in each trimester and after birth were performed. All remaining studies were singular.

**Figure 3 biomedicines-13-00626-f003:**
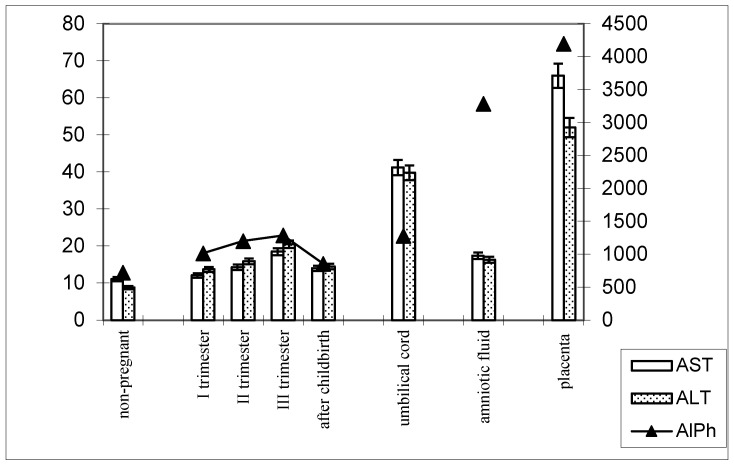
Comparative dynamics of the activities of transaminases (AST and ALT) and alkaline phosphatase in blood sera of non-pregnant and pregnant women with timely deliveries. Note: Left axis—transaminases (U/mL); right axis—alkaline phosphatase (U/mL).

**Figure 4 biomedicines-13-00626-f004:**
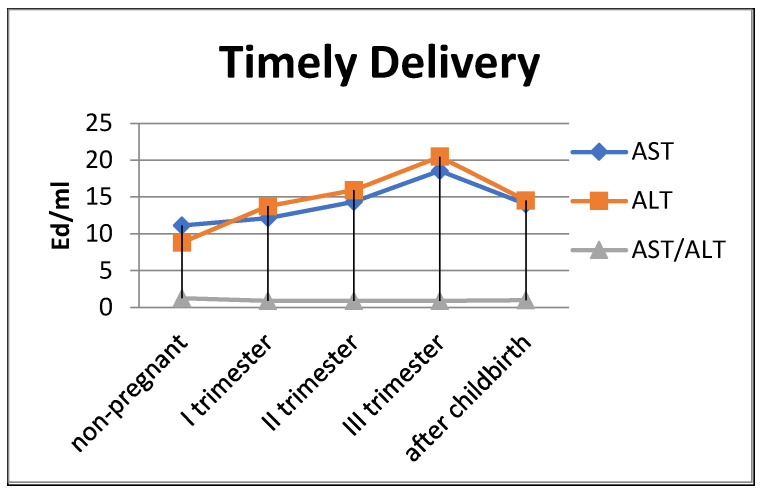
Transaminase activity of blood sera (U/mL) in the control group and pregnant women who gave birth on time, during trimesters of pregnancy, and after childbirth.

**Figure 5 biomedicines-13-00626-f005:**
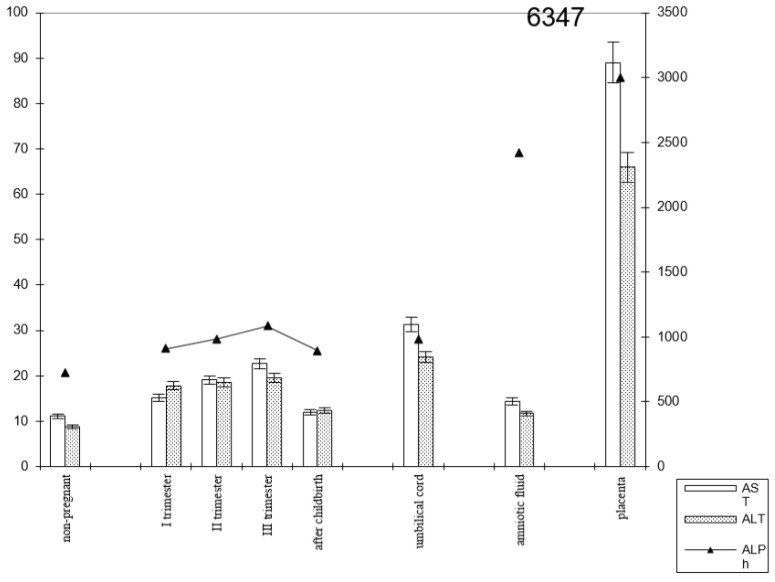
Comparative dynamics of transaminase activity (AST and ALT) and alkaline phosphatase in blood sera of non-pregnant and pregnant women with premature births. Note: 6347—AlPh level; see Figure 3.

**Figure 6 biomedicines-13-00626-f006:**
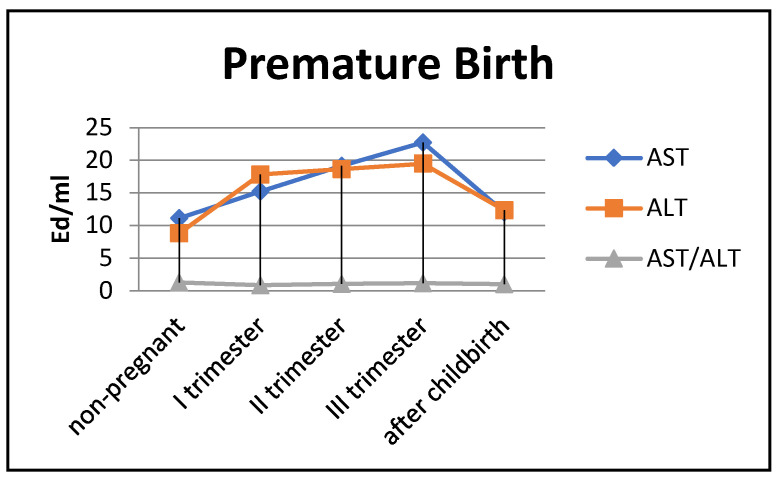
Transaminase activity in blood sera (U/mL) of the control group and pregnant women with premature deliveries in the trimesters of pregnancy and after childbirth.

**Figure 7 biomedicines-13-00626-f007:**
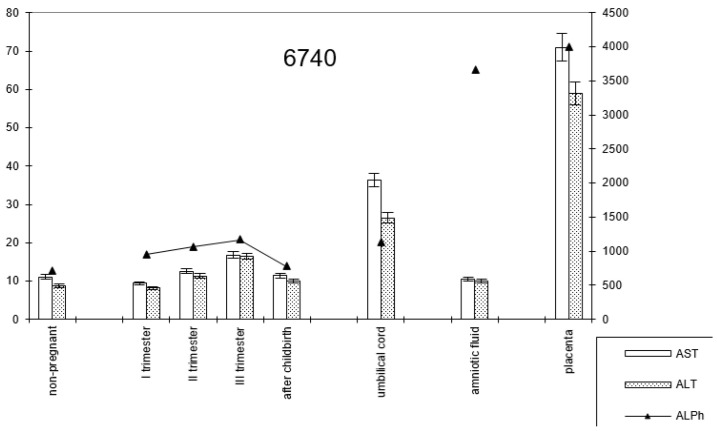
Comparative dynamics of transaminase activity (AST and ALT) and alkaline phosphatase in blood sera of non-pregnant and pregnant women with late deliveries. Note: 6740—AlPh level; see Figure 1.

**Figure 8 biomedicines-13-00626-f008:**
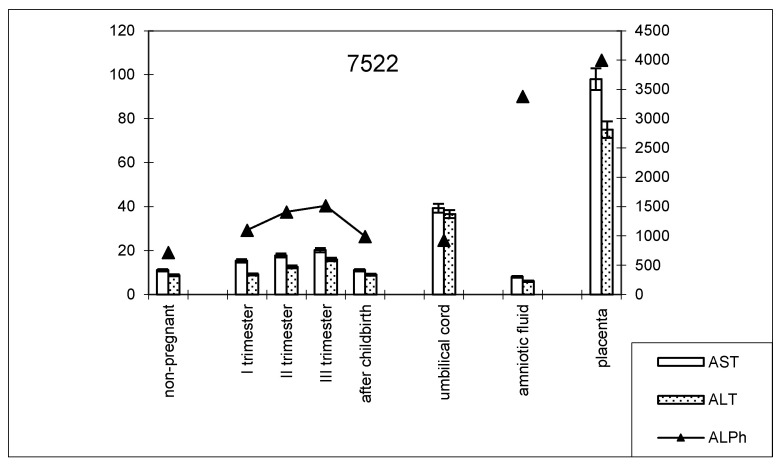
Comparative dynamics of transaminase activity (AST and ALT) and alkaline phosphatase in blood sera of non-pregnant and pregnant women with cesarean sections. Note: 7522—AlPh level; see Figure 1.

**Table 1 biomedicines-13-00626-t001:** Height and weight indicators of pregnancy and childbirth in women with different terms and types of delivery.

Parameter	Groups of Pregnant Women			
	Timely Birth (n = 86)	Premature Birth (n = 34)	Late Birth (n = 31)	Childbirth Via Cesarean Section (n = 42)
Height, sm	164.2 ± 7.9	164.6 ± 9.5	168.5 ± 1.2	166.8 ± 10.4
Pre-pregnancy body weight, kg	68.2 ± 3.6	61.8 ± 2.4	81.1 ± 5.6	71.1 ± 3.9
Total weight gain during pregnancy, kg	9.3 ± 0.4	10.3 ± 0.7	12.6 ± 0.6	11.8 ± 0.7
Weight–height index, g/sm	415.3 ± 21.3	375.4 ± 17.9	481.3 ± 29.4	426.5 ± 25.1

**Table 2 biomedicines-13-00626-t002:** Transaminase activity in blood sera (U/mL) of control group individuals and pregnant women with late deliveries in each trimester of pregnancy and after childbirth.

Parameter	Control Group	Pregnant Women with Late Childbirth			
		1st Trimester	2nd Trimester	3rd Trimester	After Childbirth
AST	11.13 ± 1.23	9.41 ± 0.86 **	12.62 ± 1.07	16.84 ± 1.22 *	11.43 ± 0.95
ALT	8.81 ± 0.72	8.24 ± 0.81	11.36 ± 0.97 **	16.51 ± 1.27 *	9.92 ± 0.89
AST/ALT	1.26 ± 0.04	1.14 ± 0.03	1.11 ± 0.03	1.02 ± 0.02 **	1.15 ± 0.03

Note: The reliability of differences for the indicators of the control group: *—*p* < 0.001; **—*p* < 0.05.

**Table 3 biomedicines-13-00626-t003:** Transaminase activity in blood sera (U/mL) of control group and pregnant women with cesarean sections in each trimester of pregnancy and after childbirth.

Parameter	Control Group	Pregnant Women with C-Section			
		1st Trimester	2nd Trimester	3rd Trimester	After Childbirth
AST	11.13 ± 1.23	15.41 ± 1.13	17.82 ± 1.27 **	20.25 ± 1.31 *	11.12 ± 1.18
ALT	8.81 ± 0.72	9.23 ± 0.85	12.64 ± 1.20 **	16.0 ± 1.28 *	9.12 ± 0.93
AST/ALT	1.26 ± 0.04	1.67 ± 0.07 **	1.41 ± 0.05	1.26 ± 0.04	1.22 ± 0.04

Note: The reliability of differences for the indicators of the control group: *—*p* < 0.001; **—*p* < 0.05.

**Table 4 biomedicines-13-00626-t004:** Alkaline phosphatase activity indices in placental homogenate in groups of pregnant women with different terms and types of delivery.

Enzyme	Groups of Pregnant Women			
	Timely Birth (n = 86)	Premature Birth (n = 34)	Late Birth (n = 31)	Childbirth by Cesarean Section (n = 42)
AlPh, U/mL	6906.2 ± 208.1	6257.4 ± 184.4 (*p* = 0.052)	7437.1 ± 212.4	7726.2 ± 216.5 (*p* = 0.068)

Note: Reliability of differences (*p*) compared with indicators in pregnant women with timely births.

**Table 5 biomedicines-13-00626-t005:** Transaminase activities in placental homogenates in groups of women depending on the timing and type of delivery.

Enzyme	Groups of Pregnant Women			
	Timely Birth (n = 86)	Premature Birth (n = 34)	Late Birth (n = 31)	Childbirth Via Cesarean Section (n = 42)
AST, U/mL	62.31 ± 4.30	41.72 ± 2.91 (*p* = 0.045)	83.44 ± 5.82 (*p* = 0.086)	76.21 ± 4.45 (*p* = 0.037)
ALT, U/mL	66.53 ± 3.72	62.80 ± 3.32	71.22 ± 4.81	53.98 ± 2.65
AST/ALT	0.94 ± 0.02	0.66 ± 0.01	1.17 ± 0.03 (*p* = 0.048)	1.41 ± 0.06 (*p* = 0.00012)

Note: Reliability of differences (*p*) compared with indicators in pregnant women with timely births.

**Table 6 biomedicines-13-00626-t006:** Transaminase activity in umbilical cord blood of women depending on timing and type of delivery.

Enzyme	Groups of Pregnant Women			
	Timely Birth (n = 86)	Premature Birth (n = 34)	Late Birth (n = 31)	Childbirth Via Cesarean Section (n = 42)
AST, U/mL	41.26 ± 3.09	31.30 ± 3.38 (*p* = 0.044)	36.31 ± 5.26	41.54 ± 3.10
ALT, U/mL	39.73 ± 3.46	24.18 ± 2.81 (*p* = 0.0001)	26.48 ± 2.52 (*p* = 0.032)	38.57 ± 2.99
AST/ALT	1.04 ± 0.02	1.29 ± 0.04 (*p* = 0.026)	1.37 ± 0.05 (*p* = 0.00012)	1.08 ± 0.02

Note: Reliability of differences (*p*) compared with indicators in pregnant women with timely births.

## Data Availability

The data for this project are confidential but may be obtained under the data use agreements of the Saint Petersburg State Pediatric Medical University, the Head of the Department of Normal Physiology. Researchers interested in accessing the data may contact Sergey Lytaev at physiology@gpmu.org. It could take some weeks (months) to negotiate the data use agreements and gain access to the data. The author will assist with any reasonable replication attempts for one year following publication.

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
