# Peer review of "The Dynamics of Transaminase and Alkaline Phosphatase Activities in the “Mother–Placenta–Fetus” Functional System"

_biomedicines, 2025, doi:10.3390/biomedicines13030626_

Round 1

Reviewer 1 Report

Comments and Suggestions for Authors

The article entitled "Dynamics of Transaminase and Alkaline Phosphatase Activity in Liquid Media of the Functional System "Mother – Placenta – 3 Fetus"" showed an interesting topic among national scientific literature. The Authors analyzed the activity of enzymes (alkaline phosphatase – AlPh and transaminases – AST, ALT) during pregnancy, providing transport and metabolic processes of the functional system "mother – placenta – fetus" and reflecting cytolysis in organs and tissues. The highest alkaline phosphatase activity was found in the placenta homogenate (6906.2 ± 208.1 U/ml) in pregnant women who delivered at term.

The Introduction and Results section have been written clearly and logically. However, the Reviewer has some concerns.

1. The Methodology Section. There is no word about the exclusion criteria. What about liver disease? Taking medicines? Alcohol habit? Was a survey with patients conducted? It should be included in this section.

2. In the Reviewers opinion it should be clearly stated in which country, what years the study was conducted. Moreover, were the patients included in the study were Caucasian? 

Author Response

Dear Reviewer! We highly appreciate your work in improving our paper. We have carefully studied your comments, changed the text with the addition of figures and tables. Below we respond to the comments step by step.

  1. The Methodology Section. There is no word about the exclusion criteria. What about liver disease? Taking medicines? Alcohol habit? Was a survey with patients conducted? It should be included in this section.

  1. In the Reviewers opinion it should be clearly stated in which country, what years the study was conducted. Moreover, were the patients included in the study were Caucasian? 

Answer: Added to Methods.

The study involved 193 pregnant women, white, european, aged 18 to 35 years in 2022-2024. They had no pathology of the gastrointestinal tract, liver, did not use medications, alcohol or nicotine. To select patients, a clinical interview and preliminary questionnaire of pregnant women were conducted to exclude pathology of organs and systems.

Reviewer 2 Report

Comments and Suggestions for Authors

This study investigated the levels of liver enzyme throughout pregnancy, using non-pregnant women as controls. The following are suggested to further improve this paper.

MAJOR

1. Figures 3 and 5 show a line graph that connects the values of control group (non-pregnant) to the values of pregnant women in various trimesters. This is misleading because the non-pregnant patients are not the same patients as those that became pregnant (unless the measurements were performed on these pregnant patients before pregnancy). 

2. Please show the exact value (up to 3 decimal places) of the p values presented in this study. Place these values within paragraphs and tables.

3. What are the factors that affect AST, ALT and AlPh levels in pregnant women? Did the investigators study the effect of age, pre-pregnancy weight and/or weight gain during pregnancy on the enzyme levels? 

4. Please expound on this statement and explain how the results of the study support this claim: "Thus, the fetal-maternal relationship is ensured by the stabilization of the uteroplacental barrier, which is determined by the level of transaminases and alkaline phosphatase in the blood serum."

5. Please improve on Figure 1 and show a much clearer diagram of the cohorts and the methods they underwent.

6. The discussion and conclusion portions of this paper require a complete overhaul. The authors need to focus on their study objectives and provide answers on those objectives in the context of relevant published information. Avoid restating the results and avoid extravagant claims based solely on the serum levels.

7. There are multiple systemic and dietary factors that can affect liver enzyme levels. How did the authors treat these factors in this study?

8. Please show separate results of the placental and umbilical cord enzyme levels. Discuss this separately as these specimens may have more infant blood in them than maternal source. Look into published normal perinatal levels to use as baseline instead of using pregnant women serum levels.

MINOR

1. The manuscript requires an overall review of grammar and spelling errors. There are also use of non-conventional terms that need to be avoided (e.g. enzymemia).

2. Figure captions should describe all symbols shown in the figure and what they represent. There are also random numbers shown in some of the figures (figures 6 and 7).

3. Be consistent with p values. It is common practice to not use p<0.05 and to show the exact values up to 3 decimal places.

Comments on the Quality of English Language

English language needs major improvement.

Author Response

Dear Reviewer! We highly appreciate your work in improving our paper. We have carefully studied your comments, changed the text with the addition of figures and tables. Below we respond to the comments step by step.

  1. Figures 3 and 5 show a line graph that connects the values of control group (non-pregnant) to the values of pregnant women in various trimesters. This is misleading because the non-pregnant patients are not the same patients as those that became pregnant (unless the measurements were performed on these pregnant patients before pregnancy). 

Answer: The enzyme levels in non-pregnant women were taken as a control to show changes in pregnant women. Non-pregnant women are a group of practically healthy women, aged 18 to 35 years, without any pathology from the gastrointestinal tract, undergoing routine screening studies.

  1. Please show the exact value (up to 3 decimal places) of the p values presented in this study. Place these values within paragraphs and tables.

Answer: Added tables 4-6.

  1. What are the factors that affect AST, ALT and AlPh levels in pregnant women? Did the investigators study the effect of age, pre-pregnancy weight and/or weight gain during pregnancy on the enzyme levels?

Answer: We did not study the effects of age, pre-pregnancy weight, and weight gain during pregnancy on enzyme levels.

  1. Please expound on this statement and explain how the results of the study support this claim: "Thus, the fetal-maternal relationship is ensured by the stabilization of the uteroplacental barrier, which is determined by the level of transaminases and alkaline phosphatase in the blood serum."

Answer: The word “stabilization” has been replaced, since we are essentially talking about the mechanisms of penetration of enzymes from the mother’s blood into the fetus’s blood.

  1. Please improve on Figure 1 and show a much clearer diagram of the cohorts and the methods they underwent.

Answer: Added description in the text and in the figure caption.

  1. The discussion and conclusion portions of this paper require a complete overhaul. The authors need to focus on their study objectives and provide answers on those objectives in the context of relevant published information. Avoid restating the results and avoid extravagant claims based solely on the serum levels.

Answer: Changes have been made. Including the addition of the original figure 1 with the relationship between organs and enzymes to the Introduction. Changes have been made to the Discussion.

  1. There are multiple systemic and dietary factors that can affect liver enzyme levels. How did the authors treat these factors in this study?

Answer: The study included women who were not on special diets that affect liver enzyme levels.

  1. Please show separate results of the placental and umbilical cord enzyme levels. Discuss this separately as these specimens may have more infant blood in them than maternal source. Look into published normal perinatal levels to use as baseline instead of using pregnant women serum levels.

Answer: Added new results – tables 4-6 with description.

Minor

  1. The manuscript requires an overall review of grammar and spelling errors. There are also use of non-conventional terms that need to be avoided (e.g. enzymemia).

Answer: Changed to «enzyme level in the blood».

  1. Figure captions should describe all symbols shown in the figure and what they represent. There are also random numbers shown in some of the figures (figures 6 and 7).

Answer: The high numbers are not accidental. This is the alkaline phosphatase level. Since its dimension is thousands of times greater than ALT - AST, it is marked with a separate number. The transcript is added to the signature.

  1. Be consistent with p values. It is common practice to not use p<0.05 and to show the exact values up to 3 decimal places.

Answer: Added in new tables. Although the p<0.05 (0.01, 0.001) value is also used in scientific articles.

Reviewer 3 Report

Comments and Suggestions for Authors

The manuscript entitled "Dynamics of Transaminase and Alkaline Phosphatase Activity in Liquid Media of the Functional System "Mother – Placenta – Fetus" is focused on the roles of ALT, AST and ALPh and their roles in pregnancy and communication between mother and fetus. The quality of English is poor, some sentences is difficult to understand and I recommend for the authors to perform proofreading. Moreover, I have multiple questions for the authors, therefore I put the "Major revision required".

Major

1. The number of investigated patients is small. Thus, in similar work more than 34.000 of women were investigated  (DOI: 10.1080/14767058.2021.1892633).

2. Introduction section is poor written .Authors should more focus the reader attention in the role of ALT, AST, ALPh in the communication of mother and fetus and their role in placenta functioning. It is important to provide in Introduction section some figure illustrating communications between mother - placenta - fetus in context of ALT, AST and ALPh. In the initials of Introduction section provide more information about the role of these enzymes. Also add brief explanation of experiments performed in the current work. 

3. Statistics section. Why you used Mann-Whitney criteria? How did you checked that your distribution is not normal? You performed multiple comparisons? Did you used Bonferroni correction?

4. Are statistically significant differences of ALT, AST, ALPh and de Ritis coef when comparing one group of patients at the diferent time points (for example, females with term normal delivery 1st trimester, 2nd trimester, etc)?

5. If there are some correlations between ALT, AST, ALPh levels and the state of the newborn (weight, height, Apgar score, other markers)?

6. Figure 2. 1, 2 and 3 trimesters are long periods. At which time samples were obtained?

7. Lines 246-266. Why this part is in the section Results? It should be in Discussion/Introduction.

8. Which method you used to measure levels of enzymes? Describe it in details.

Minor

1. Provide full variants of the following abbreviations ALPh, ALT, AST

2. LINES 109-113. How these two sentences are related one to other?

Comments on the Quality of English Language

1. Line 63 "needs require" check

2. Line 139 named after?

3. Line 161 opening of what?

Author Response

Dear Reviewer! We highly appreciate your work in improving our paper. We have carefully studied your comments, changed the text with the addition of figures and tables. Below we respond to the comments step by step.

  1. The number of investigated patients is small. Thus, in similar work more than 34.000 of women were investigated  (DOI: 10.1080/14767058.2021.1892633).

Answer: This work did not require such a large number, since significant differences were obtained with the sample sizes used. We examined more than 200 women of similar age and health status. Our main goal is to evaluate enzyme distribution mechanisms. In works with epidemiological tasks or GCP, the number is agreed upon in advance.

  1. Introduction section is poor written .Authors should more focus the reader attention in the role of ALT, AST, ALPh in the communication of mother and fetus and their role in placenta functioning. It is important to provide in Introduction section some figure illustrating communications between mother - placenta - fetus in context of ALT, AST and ALPh. In the initials of Introduction section provide more information about the role of these enzymes. Also add brief explanation of experiments performed in the current work. 

Answer: Added original Figure 1 to Introduction. Transaminases in their optimal ratio regulate metabolic processes. AST is a metabolism integrator, a marker of the central catabolism zone, an indicator of lipid peroxidation (LPO) of cytolysis. ALT is a component of the glucose-alanine shunt, a marker of the peripheral catabolism zone. The ratio of AST to ALT (De Ritis coefficient) is an indicator of adaptation of metabolic flows. These enzymes and alkaline phosphatase provide metabolic shifts. The research part is described in more detail in Methods.

  1. Statistics section. Why you used Mann-Whitney criteria? How did you checked that your distribution is not normal? You performed multiple comparisons? Did you used Bonferroni correction?

Answer: Added to paper.

First, the distribution was checked for normality. We made sure that the distribution was not normal, so we used the nonparametric Mann-Whitney test. Differences between the groups in the level of the studied characteristics were assessed using the nonparametric Mann-Whitney method using the statistical package SPSS 11.0. Differences were considered significant at an error probability of p < 0.05, and significance levels were also divided into p < 0.01 and p < 0.001. We found a negative correlation between the activity of alkaline phosphatase in the umbilical cord blood of women with premature birth and the level of this enzyme in the child's urine on the first day of life (r = - 0.51 ± 0.11; p = 0.056). There was a strong association between AST and ALT activity in cord blood in women who delivered at term (r=+0.86±0.15; p=0.001) and a moderate association in women with preterm delivery (r=+0.53±0.12; p=0.088).

  1. Are statistically significant differences of ALT, AST, ALPh and de Ritis coef when comparing one group of patients at the different time points (for example, females with term normal delivery 1st trimester, 2nd trimester, etc)?

Answer: Added to paper.

The AST activity in the placental homogenate of women who delivered at term was 62.31±4.30 U/ml. In premature births, the enzyme activity was 1.5 times (p=0.045) lower than in full-term births. In late births and births by cesarean section, an increase in AST activity was observed compared to full-term births (by 1.3 times (p=0.086) and 1.2 times (p=0.037), respectively). There was a weak relationship between the AST activity in the placental homogenate of women who delivered at term and by cesarean section (r=+0.24±0.07; p=0.054). With regard to ALT activity in this biofluid, the same pattern of changes was revealed in the studied groups of women as for AST. The difference was a decrease in ALT activity in women who gave birth by caesarean section (1.2 times compared to women who gave birth at term).

  1. If there are some correlations between ALT, AST, ALPh levels and the state of the newborn (weight, height, Apgar score, other markers)?

Answer: No comparative analysis was performed between the levels of ALT, AST, and ALP in women and the condition of the newborn.

  1. Figure 2. 1, 2 and 3 trimesters are long periods. At which time samples were obtained?

Answer: The activity of alkaline phosphatase and transaminases (AST, ALT) was determined in the blood serum of pregnant women at 12-13, 25-26 and 39-40 weeks of pregnancy, and once in individuals in the control group.

  1. Lines 246-266. Why this part is in the section Results? It should be in Discussion/Introduction.

Answer: Moved to Discussion.

  1. Which method you used to measure levels of enzymes? Describe it in details.

Answer: Added to manuscript.

The alkaline phosphatase activity in blood serum, umbilical cord blood, and placenta homogenate was determined using a standardized method with reagent kits manufactured by Lahema Diagnosticum (Czech Republic). The principle is that alkaline phosphatase breaks down 4-nitrophenyl phosphate in N-methyl-glucamine buffer to form 4-nitrophenol and phosphate. The amount of 4-nitrophenol, which was determined photometrically by the constant time method after stopping the enzymatic reaction with an alkaline phosphatase inhibitor that blocks the active center of the enzyme, is a measure of the catalytic concentration of the enzyme.

To determine the activity of AST and ALT in blood serum, umbilical cord blood, and placenta homogenate, we used the colorimetric dinitrophenylhydrazine method according to Reitman and Frenkel. The principle of the method is that AST catalyzes the reaction between α-aspartate and 2-oxoglutarate, as a result of which they are converted into α-glutamate and oxaloacetate. The determination is based on measuring the optical density of 2-oxoglutaric and pyruvic acid hydrazones in an alkaline medium. ALT catalyzes the reaction between α-alanine and 2-oxoglutarate, as a result of which they are converted into α-glutamate and pyruvic acid salt. The determination is based on measuring the optical concentration of 2-oxoglutaric and pyruvic acid hydrazones in an alkaline medium.

Minor

  1. Provide full variants of the following abbreviations ALPh, ALT, AST

ALPh - Alkaline phosphаtase

ALT – Alanine aminotransferase

AST - Aspartate aminotransferase

Answer: Added to abstract.

  1. LINES 109-113. How these two sentences are related one to other?

From the standpoint of morphofunctional sciences, blood is a unique tissue that has the ability to regulate the flow of metabolites into the physiological systems of the body, being a liquid medium for the functioning of a number of enzymes.

Enzymes can be classified not only as indicators of the state or damage of organs, but also as biologically active substances that perform a certain physiological role.

Answer: On paper it's one sentence - From the standpoint of morpho-functional sciences, blood is a unique tissue that has the ability to regulate the flow of metabolites into the physiological systems of the body, being a liquid medium for the functioning of a number of enzymes that can be classified not only as indicators of the state or damage of organs, but also as biologically active substances that perform a certain physiological role.

Answer: Blood is one of the body's liquid spaces, and enzymes, which in clinical medicine are better known as indicators of pathological processes, we consider from the point of view of regulating metabolic processes in different liquid spaces.

Comments on the Quality of English Language

Answer: If the paper is going to print, we will order MDPI editing.

Reviewer 4 Report

Comments and Suggestions for Authors

1. Regarding the ethical issues, the pregnant women (both control and research groups) are vulnerable group. Did the authors take any more extra cautions for taking in the process of taking informed consent from the participants as well as form her partner? Please describe more details if the authors did it.

2. Regarding the enrollment, how did the authors recruit the pregnant women (Eg at first antenatal visit?). Then the authors described that the authors took peripheral blood from the pregnant women during first, second and third trimesters. Did the authors determine any specific weeks such as 12 weeks for first trimester, 25 weeks for second trimester? Or the authors allow no specific weeks or date and allow if it was within trimester? Please clearly describe it.

3. Liver enzymes can affect the participants have underlying hepatitis infection. Did the authors check hepatitis viruses (HBV, HCV, HDV, etc) for the enrollment of the participants in the study?

4. The total number of pregnant can vary the enzyme (Eg, multipara, primipara, etc). Did the authors determine the number of pregnancy for the  enrollment in your study.

5. Regarding the control group (non-pregnant women), how many times did the authors take venous blood rom the participants? Please clearly describe at the revised manuscript.

Author Response

Dear Reviewer! We highly appreciate your work in improving our paper. We have carefully studied your comments, changed the text with the addition of figures and tables. Below we respond to the comments step by step.

  1. Regarding the ethical issues, the pregnant women (both control and research groups) are vulnerable group. Did the authors take any more extra cautions for taking in the process of taking informed consent from the participants as well as form her partner? Please describe more details if the authors did it.

Answer: We took informed consent approved by the order of the Ministry of Health. When submitting the paper, the editorial board asks us for forms. It sets out in detail the objectives of the research and possible risks. We spent more time explaining before signing.

  1. Regarding the enrollment, how did the authors recruit the pregnant women (Eg at first antenatal visit?). Then the authors described that the authors took peripheral blood from the pregnant women during first, second and third trimesters. Did the authors determine any specific weeks such as 12 weeks for first trimester, 25 weeks for second trimester? Or the authors allow no specific weeks or date and allow if it was within trimester? Please clearly describe it.

Answer: The activity of alkaline phosphatase and transaminases (AST, ALT) was determined in the blood serum of pregnant women at 12-13, 25-26 and 39-40 weeks of pregnancy, and once in the control group.

  1. Liver enzymes can affect the participants have underlying hepatitis infection. Did the authors check hepatitis viruses (HBV, HCV, HDV, etc) for the enrollment of the participants in the study?

Answer: The absence of hepatitis viruses (HBV, HCV, HDV, etc.) was checked upon inclusion of participants in the study - this is a standard test for pregnant women.

  1. The total number of pregnant can vary the enzyme (Eg, multipara, primipara, etc). Did the authors determine the number of pregnancy for the  enrollment in your study.

Answer: Only first-time pregnant women participated in the study.

  1. Regarding the control group (non-pregnant women), how many times did the authors take venous blood rom the participants? Please clearly describe at the revised manuscript.

 Answer: The activity of alkaline phosphatase and transaminases (AST, ALT) was determined in the blood serum of non-pregnant women (control group) - once.

Reviewer 5 Report

Comments and Suggestions for Authors

This study analyzed the activity of alkaline phosphatase (AlPh) and transaminases (AST, ALT) in various biological media (blood serum, amniotic fluid, cord blood, placenta homogenate) across pregnancy trimesters and postpartum periods, focusing on different delivery types (term, premature, late, cesarean). The de Ritis coefficient (AST/ALT) and enzyme activities were determined using standard methods. Results showed the highest AlPh activity in placental homogenate at term and elevated transaminase activity in cord blood and placenta in premature and late deliveries. These enzymes play a crucial role in metabolic transport and maintaining enzyme composition in blood plasma. While the manuscript is well-structured and supported by rigorously designed experiments, several areas require improvement to enhance its clarity and impact:

1.     The abstract should be concise.

2.     Revise keywords to avoid redundancy with the title.

3.     The objective of the study could be more clearly defined. The abstract and introduction suggest multiple aims, making it difficult to discern the primary focus.

4.     While the sample size is presented, there is no discussion of how it was determined.

5.   More comprehensive information about how normality was tested is needed.

Author Response

Dear Reviewer! We highly appreciate your work in improving our paper. We have carefully studied your comments, changed the text with the addition of figures and tables. Below we respond to the comments step by step.

  1. The abstract should be concise.

Answer: Abstract was reduced.

  1. Revise keywords to avoid redundancy with the title.

        Answer: Edited the title (shortened) and keywords.

  1. The objective of the study could be more clearly defined. The abstract and introduction suggest multiple aims, making it difficult to discern the primary focus.

     Answer: The wording of the goal has been edited.

  1. While the sample size is presented, there is no discussion of how it was determined.

More comprehensive information about how normality was tested is needed.

Answer: First, the distribution was checked for normality. When the fact that the distribution was not normal was confirmed, the nonparametric Mann-Whitney criterion was used. Differences between groups in the level of the studied characteristics were estimated by the nonparametric Mann-Whitney method using the statistical package SPSS 11.0. Differences were considered significant at an error probability of p<0.05, and significance levels of p<0.01 and p<0.001 were also distinguished. Statistical processing of the results was performed using Microsoft Excel 2003 spreadsheets, SPSS 23.0 and Primer of biostatistics 4.03 programs.

Round 2

Reviewer 2 Report

Comments and Suggestions for Authors

There has been no improvement in the discussion and conclusion of this manuscript. The results are presented haphazardly and are difficult to read and understand. The aim of this paper is not clearly stated nor achieved by the results.

Comments on the Quality of English Language

English language use requires major revision.

Author Response

Dear Reviewer! Thank you for your efforts to improve our manuscript. We respond to these comments as follows:

Q: There has been no improvement in the discussion and conclusion of this manuscript. The results are presented haphazardly and are difficult to read and understand. The aim of this paper is not clearly stated nor achieved by the results.

A: We shortened and specified the goal. We reworked the conclusion for a specific goal.

Q: English language use requires major revision.

A: The manuscript has been rewritten using the MDPI English language correction system.

Reviewer 3 Report

Comments and Suggestions for Authors

Authors significantly improved their manuscript, however is still far from that to be ideal.

1. Statistics section. You should indicated how did you checked normality of your data. Which parameter did you used?

2. In the Introduction section you added figure. This figure may present in the introduction section. However, during the first round of review I asked you to make figure which illustrates interaction between mother-fetus-placenta in context of ALT, AST, etc. It is still not performed.

3.  "If there are some correlations between ALT, AST, ALPh levels and the state of the newborn (weight, height, Apgar score, other markers)?

Answer: No comparative analysis was performed between the levels of ALT, AST, and ALP in women and the condition of the newborn."

You should perform this analysis. You have all necessary data for this.

Author Response

Dear Reviewer! Thank you for your efforts to improve our manuscript. We respond to these comments as follows:

Q: Statistics section. You should indicated how did you checked normality of your data. Which parameter did you used?

A: Added to text. Normality was determined using the Pearson criterion. Average values ​​(M), their standard error (m) and standard deviation (s) were calculated. The dependence between the features was assessed using the pair correlation coefficient (r), its error (mr) and the significance level of differences (according to Student's t-criterion). The dependence was considered strong when êr ê > 0.7, average, if the modulus of the pair correlation value lies within 0.3-0.7. If the correlation value was found to be less than the modal value of 0.3, it was considered weak.

Q: In the Introduction section you added figure. This figure may present in the introduction section. However, during the first round of review I asked you to make figure which illustrates interaction between mother-fetus-placenta in context of ALT, AST, etc. It is still not performed.

A: Improved the figure. Added to the text, as well as the fetus with the placenta and alkaline phosphatase. The sources of alkaline phosphatase in the blood can be the liver, gastrointestinal tract, fallopian tubes, bones and organs with transport and exchange functions.

Q: "If there are some correlations between ALT, AST, ALPh levels and the state of the newborn (weight, height, Apgar score, other markers)? You should perform this analysis. You have all necessary data for this.

A: Thank you for your remark. We have such information. However, including this data would significantly increase the volume of the manuscript and would go beyond the stated purpose. We will take this into account in the next publication.

Reviewer 5 Report

Comments and Suggestions for Authors

Now, after addressing all the comments, the quality of the manuscript potentially improved. I recommend it for publication in its present form.

Author Response

Dear Reviewer! Thank you for your efforts to improve our manuscript.